# Transition Metal Containing Particulate Matter Promotes Th1 and Th17 Inflammatory Response by Monocyte Activation in Organic and Inorganic Compounds Dependent Manner

**DOI:** 10.3390/ijerph17041227

**Published:** 2020-02-14

**Authors:** Adrianna Gałuszka, Małgorzata Stec, Kazimierz Węglarczyk, Anna Kluczewska, Maciej Siedlar, Jarek Baran

**Affiliations:** Department of Clinical Immunology, Institute of Pediatrics, Jagiellonian University Medical College, Wielicka Street 265, 30-663 Cracow, Poland; adrianna.galuszka@doctoral.uj.edu.pl (A.G.); stecmalgorzata@gmail.com (M.S.); kazimierz.weglarczyk@uj.edu.pl (K.W.); ania.gruca@uj.edu.pl (A.K.); misiedla@cyf-kr.edu.pl (M.S.)

**Keywords:** transition metal containing particulate matter, air pollution, CD4+ T cells, monocytes, cytokines, endotoxin

## Abstract

In recent years, a significant increase in the frequency of disorders caused by air pollutants has been observed. Here we asked whether transition metal-containing particulate matter (TMCPM), a component of air pollution, has an effect on the activity of human CD4+ T cell subsets (Th1, Th2, Th17, and Treg). Peripheral blood mononuclear cells (PBMC) from healthy donors were cultured with or without NIST (SRM 1648a—standard urban particulate matter purchased from the National Institute for Standards and Technology) and LAP (SRM 1648a particulate matter treated within 120 min with cold oxygen plasma) preparations of TMCPM, differing in organic compounds content. Data show that TMCPM treatment increased the level of CD4+ cells positive for IFN-γ and IL-17A, specific for Th1 and Th17 cells, respectively. Moreover, a substantial decrease in frequency of Foxp3 positive CD4+ cells was observed in parallel. This effect was more pronounced for NIST particles, containing more organic components, including endotoxin (LPS - lipopolysaccharide) and required the presence of monocytes. Inactivation of LPS by treatment of TMCPM with polymyxin B reduced the inflammatory response of monocytes and Th subsets but did not abolish this activity, suggesting a role of their inorganic components. In conclusion, treatment of human PBMC with TMCPM skews the balance of Th1/Th2 and Treg/Th17 cells, promoting polarization of CD4+ T cells into Th1 and Th17 subsets. This phenomenon requires activation of monocytes and depends on the organic and inorganic fractions, including endotoxin content in TMCPM, as significantly higher inflammatory response was observed for the NIST comparing to LAP. This observation may shed a new light on the role of TMCPM in development and exacerbation of allergies, inflammatory, and autoimmune disorders.

## 1. Introduction

In recent years, an increase in air pollution has seriously impacted human’s health [1,2]. It has been accepted that air pollutants, specifically respirable particulate matter (PM), in association with genetic predispositions, environmental, and epigenetic factors promote initiation of inflammatory reactions and development of allergy, inflammatory or autoimmune disorders [3,4,5,6,7]. Furthermore, the exposure to PM is associated with increased risk of pulmonary and cardiovascular disease-related morbidity, and mortality [8,9,10,11]. TMCPM (transition metal containing particulate matter) consists of the fine, ultrafine, and nanoparticles containing metals, such as iron (Fe), zinc (Zn), copper (Cu), chromium (Cr), nickel (Ni), manganese (Mn), lead (Pb), and cadmium (Cd) [12,13,14]. Although transition metals are required cofactors for many proteins that are critical for life, inhaled TMCPM can deposit in different compartments in the respiratory tract and interact with epithelial and resident immune cells. Usually, they reach the lower respiratory tract where they are mainly phagocytosed by alveolar macrophages [15]. Smaller PM can penetrate the lower respiratory tract more efficiently and therefore are more harmful to human’s health. Part of them can also translocate from the lung to the blood stream and interact with circulating leukocytes [16], including monocytes, which play a crucial role in the early phases of the immune response, phagocytosing foreign material, presenting antigens to CD4+ T cells and inducing antigen-specific - adaptive immune response [17]. 

The human immune system is a sensitive target for air pollution and the optimal balance between activation of different cells, including Th cell subsets (Th1/Th2 and Th17/Treg) is a key element responsible for its homeostatic maintenance. Dysregulation of the Th1/Th2 ratio leads to excessive Th1 or Th2 cell activation, resulting in the development of inflammatory or autoimmune diseases often associated with accumulation of Th1 cells or the induction of allergic diseases due to the accumulation of Th2 cells, respectively. On the other hand, an elevated level of Th17 cells and peripheral Th17/Treg imbalance is associated with the development of autoimmune diseases and may also contribute to the pathogenesis of classically recognized Th2-mediated allergic disorders [18]. Although, the effect of the immune cell exposure to PM has been well documented in many animal models [19,20], the influence of TMCPM on human CD4+ T cells has been poorly defined. In this study, we investigated the effect of TMCPM on the activity of peripheral blood CD4+ T cell subsets (Th), determined by the expression of intracellular proteins, namely IFN-γ, IL-4, IL-17A, and Foxp3, (markers for Th1, Th2, Th17, and Treg cells, respectively), in vitro after exposure of PBMC to TMCPM preparations, differing in the content of organic compounds. This work is a part of the APARIC project (Air Pollution versus Autoimmunity: Role of multiphase aqueous Inorganic Chemistry) aimed to study the influence of transition metal compounds as components of air pollutants on the induction of autoimmune disorders [21].

## 2. Materials and Methods

### 2.1. TMCPM Preparation

Two different preparations of TMCPM (transition metal containing particulate matter) were used in the study: SRM 1648a—standard urban particulate matter (purchased from the National Institute for Standards and Technology, Gaithersburg, MD, USA), designated as NIST, which is a conglomeration of fine and ultrafine particles (mean particle diameter 5.85 μm), containing ca. 13% of carbon, including 10.5% of organic carbon [22] and LAP-SRM 1648a particulate matter treated within 120 min with cold oxygen plasma for the removal of organic compounds from the reference material. Samples of LAP, with removed organic content (containing less than 2% of organic carbon, and less than 1% of nitrogen) were prepared in the Department of Inorganic Chemistry, Jagiellonian University, Krakow, Poland, as described previously [23]. Both preparations of TMCPM were weighted on a high precision microbalance and freshly suspended in RPMI 1640 medium (Corning, Manassas, VA, USA) in sterile conditions. Three final concentrations of TMCPM were experimentally established as non-cytotoxic for PBMC (1 µg/mL, 10 µg/mL, 100 µg/mL) and used in further experiments.

### 2.2. Cell Isolation

ACDA-treated blood from healthy donors was purchased from the Regional Center of Blood Donation and Blood Therapy in Krakow, Poland. Peripheral blood mononuclear cells (PBMC) were isolated by standard Pancoll human (Panbiotech, Aidenbach, Germany) density gradient centrifugation, washed and resuspended in RPMI 1640 medium (Corning), containing 10% heat-inactivated fetal bovine serum (EURx, Gdańsk, Poland). Monocytes were separated from PBMCs by counter-current centrifugal elutriation (JE-6B elutriation system equipped with a 5-mL Sanderson separation chamber; Beckmann-Coulter, Palo Alto, CA, USA), as described previously [24]. Cells were washed, resuspended in RPMI 1640 medium supplemented with 2 mM of L-glutamine, 5% heat-inactivated fetal bovine serum (EURx), and 25 µg/mL gentamycin (Sigma, St. Louis, MO, USA) (complete medium) and kept in an ice bath until used. Purity of isolated monocytes was checked by flow cytometry using anti-CD14 antibody (BD Biosciences, Pharmingen, San Diego, CA, USA) and did not drop below 90%. For some experiments, population of lymphocytes containing c.a. 80% of CD3-positive cells was isolated by counter-current centrifugal elutriation from PBMCs.

### 2.3. Cell Culture and Immunostaining for Intracellular Proteins

PBMC were cultured at the density of 1 × 10^6^/mL in ultra-low-attachment tubes (Corning) in complete RPMI 1640 medium, with or without TMCPM (LAP and NIST), at three concentration 1 µg/mL, 10 µg/mL, and 100 µg/mL. Cells were kept at 37 °C, 5% CO_2_ in a humidified atmosphere. As a positive control, cells were stimulated with 50 ng/mL PMA (phorbol 12-myristate 13-acetate; Sigma) and 100 ng/mL of ionomycin (Sigma). In some experiments, in addition to PBMC, the lymphocyte population was enriched in T cells (lymphocyte fractions from counter-current centrifugal elutriation) or such T cells were added to monocytes (10:1 ratio), exposed to TMCPM pre-treated or not with Polymyxin B (InvivoGen, San Diego, CA, USA) at the concentration 100 µg/mL were used. Cells were treated and cultured in a similar manner as PBMC.

For intracellular detection of IFN-γ, IL-4, IL-17A, and FoxP3, to inhibit protein secretion, 2 µM Golgi Stop (containing monensin; BD Biosciences) was added at the beginning of culture, as was described previously [25]. After 5 h of culture (according to manufacturer’s instructions) the cells were harvested, washed once in ice-cold PBS (phosphate-buffered saline; Corning) with 5% heat-inactivated fetal bovine serum (EURx) and stained with the cocktail of fluorescently conjugated monoclonal antibodies, specific for human CD4+, IFN-γ (for Th1), IL-4 (for Th2), IL-17A (for Th17), and FoxP3 (for Treg), according to manufacturer instructions (Human Th1/Th2/Th17 and Human Th17/Treg Phenotyping Kits, BD Biosciences). Expression of intracellular proteins (IFN-γ, IL-4, IL-17A, and FoxP3) was measured by 10-colour FACS Canto flow cytometer (BD Biosciences, Immunocytometry Systems, San Jose, CA, USA) and analyzed using BD FACSDiva software version 8.0.1 (BD Biosciences). Typically, 10,000 events for CD4+ cells were recorded. Percent value of the cells positive for IFN-γ, IL-4, IL-17A, and FoxP3 were quantified as the frequency measure of Th1, Th2, Th17, and Treg cells, respectively.

### 2.4. Cytokine Detection by CBA

Monocytes at the density of 3 × 10^5^ were cultured at 37 °C in a humidified 5% CO_2_ atmosphere in Ultra Low Attachment tubes in complete medium with or without TMCPM (LAP and NIST) at the concentration 100 µg/mL. Monocytes stimulated with 400 U/mL of human recombinant IFN-γ (Sigma) and 100 ng/mL of LPS (lipopolysaccharide) from Salmonella abortus equi (Sigma) were cultured simultaneously as a positive control. Supernatants from monocyte cultures were collected and concentration of TNF-α was determined by Cytokine bead array (CBA, Human Inflammatory Cytokine Kit; BD Biosciences), according to manufacturer’s instructions. Typically, data from 3600 events for beads were acquired by a 10-colour FACSCanto flow cytometer (BD Biosciences) and analyzed using FCAP Array v. 3.0 software (BD Biosciences). 

### 2.5. Endotoxin Determination

Endotoxin (LPS) content in TMCPM (NIST and LAP) preparations and TMCPM treated with Polymyxin B (InvivoGen) (100 µg/mL; 15 min room temp) was measured by a quantitative ELISA-like heterogeneous enzymatic assay, using EndoLISA kit, according to manufacturer’s instructions (Hyglos GmbH, München, Germany). TMCPM samples were resuspended in sterile PBS (Corning) and analyses were performed at a temperature of 37 °C with automated microplate reader Infinite 200 Pro (Tecan, Männedorf, Switzerland). Endotoxin from Escherichia coli B0:55 (Sigma) was used as a standard. Measurements were repeated if the difference between sample duplicates exceeded 10%. Endotoxin concentrations were expressed as EU/mL, based on the EndoLISA standard curve (four-parameter logistic non-linear regression model) required by the test.

### 2.6. Statistics

Statistical analysis was performed by one-way ANOVA and t tests using PRISM GraphPad 6.01 software (GraphPad Software Inc., San Diego, CA, USA). Results were expressed as the mean ± standard error of measurement (SEM). Statistical significance of differences between the groups was considered at the following *p* values: * *p* < 0.01, ** *p* < 0.01, *** *p* < 0,001, **** *p* < 0.0001. Statistically significant differences in treated groups compared to controls were determined using the Dunnett test. 

## 3. Results

### 3.1. TMCPM Reduces the Expression Level of CD4 on the Surface of T Cells

Effect of TMCPM on the activity of human CD4+ T cell subsets was investigated after five hours of healthy donor PBMCs exposure to NIST or LAP particles. The concentrations of NIST and LAP used in the study were established experimentally in a range from 1–100 µg/mL, as used at higher concentrations were cytotoxic (data not shown), confirming previous reports [22]. At first, we checked weather TMCPM may affect the proportion of CD4+ T cells and the expression of CD4 on their surface, as was documented for PMA and ionomycin [25,26,27], potent activators of T cells. For this purpose, the population of lymphocytes was gated according to FSC/SSC parameters during FACS analysis of PBMC and the expression of CD4 antigen (mean fluorescence intensity), as well as the percentage of CD4+ cells, was analyzed. The results presented in Figure 1 show that the expression of the CD4 antigen on the surface of T lymphocytes decreased after cells exposition to TMCPM, similarly to the group of cells stimulated with PMA + ionomycin (Figure 1A). This effect was dose-dependent (data not shown), being most pronounced at the highest concentration (100 µg/mL) of NIST or LAP. However, the exposure of PBMC to TMCPM did not significantly change the frequency (percent) of CD4+ T cells (Figure 1B).

### 3.2. TMCPM Activates CD4+ T Lymphocytes Depending on the Organic Compounds Content

Population of CD4+ T cells is composed of several subsets, mainly Th1, Th2, Th17, and Treg. These populations were identified after the exposure of PBMC to TMCPM, according to the expression of intracellular proteins, characteristic for each subset of IFN-γ, IL-4, IL-17A, and FoxP3, respectively. The exposure of PBMC to NIST or LAP caused an increase in the proportion of cells positive for IFN-γ and IL-17A (Figure 2).

This effect was most pronounced for the highest concentration of NIST and LAP, when compared to the unstimulated control group. Additionally, LAP used at the highest concentration did induce a considerable percentage of IL-17A-positive CD4+ T cells, however, it did not reach such a high-level (Figure 2). This observation suggests that the effect of TMCPM was strongly dependent on the organic fraction component in the particle preparation, as significantly stronger immune response was detected with the use of NIST. At this time point, no IL-4-producing cells was detected after exposure of PBMC to TMCPM (Figure 2 and Figure 3A), while stimulation of PBMC with NIST or LAP induced a dose dependent decrease in the frequency of regulatory T cells, albeit with no statistical significance (data not shown). 

### 3.3. TMCPM Exposure Is Skewing the Balance of Th1/Th2 and Treg/Th17 Subsets

It is accepted that the ratio between Th1/Th2 and Treg/Th17 subsets correlates with the functional status of the immune system [28,29,30,31]. In our study, we compared these ratios in the groups of cells with and without exposure to TMCPM. Figure 3B. shows a significant increase in the Th1/Th2 ratio after cell exposure to highest concentration of NIST, being highly skewed into Th1 pattern. The exposure of PBMC to LAP at highest concentration caused also an increase in Th1/Th2 ratio, albeit less pronounced. Moreover, the cells exposed to NIST or LAP showed markedly lower Treg/Th17 ratio in comparison to control, unexposed group. This was most pronounced for NIST, regardless of the particle concentration used (Figure 3C).

Taken together, exposure of PBMC to TMCPM (NIST or LAP) caused an increase in the frequency of Th1 and Th17 cells, indicating changes in the polarization of the immune response. 

### 3.4. Monocytes Activation Is Required for TMCPM Induced Polarization of Th1 and Th17 Immune Response

To answer whether T cells can directly respond to TMCPM, fractions containing lymphocytes were collected during counter-current elutriation and such cell population, composed mainly of T lymphocytes (80 to 90% of CD3-positive cells) [31], was incubated with the selected concentrations of TMCPM. Thereafter intracellular expression of IFN-γ, IL-4, IL-17A, and FoxP3 was detected in CD4+ cells by flow cytometry and Th1/Th2 and Treg/Th17 ratios were calculated. The obtained data (Figure 4) show that the expression of intracellular proteins, characteristic for each of CD4+ subset, did not change significantly after TMCPM treatment, documenting that TMCPM did not directly affect the composition of CD4+ T cells. This observation suggests that monocytes play a crucial role in the induction of Th1 and Th17 immune response after TMCPM treatment.

With this in mind, in the next set of experiments, purified monocytes were incubated with TMCPM in a similar manner, and production of pro-inflammatory cytokines TNF-α, IL-6, and IL-8, was detected in the culture supernatants by CBA. The obtained data (Figure 5) show that monocytes directly respond to TMCPM by production of cytokines, including TNF-α, IL-6, and IL-8. 

This effect was most significant for NIST particles, while, in the case of LAP, it was observed only at the highest concentration, proving the importance of the organic compounds content in TMCPM. The critical role of monocytes in the induction of Th1 and Th17 immune response after TMCPM challenge was further confirmed by co-incubation of purified monocytes exposed to TMCPM with lymphocyte population enriched in T cells. In this setting, polarization of CD4+ T cells into Th1 and Th17 by TMCPM exposure has been restored (Figure 6).

### 3.5. Endotoxin Content Is Partially Responsible for Pro-Inflammatory Potency of TMCPM

The main organic candidate responsible for activation of human monocytes for pro-inflammatory activity of TMCPM is endotoxin (LPS). In this regard, in the next step we performed analysis of the endotoxin concentration in the suspensions of NIST and LAP particles. The obtained data (Table 1) revealed that both preparations of TMCPM contain in their organic fraction also endotoxin 1.75 EU/mL and 0.40 EU/mL (at the highest dose) for NIST and LAP, respectively. However, pre-treatment of TMCPM with polymyxin B decreased endotoxin content to 0.10 EU/mL in NIST and to 0.00 EU/mL in LAP.

TMCPM samples were resuspended in sterile PBS and endotoxin (LPS) concentration in NIST and LAP preparations was measured by a quantitative ELISA-like heterogeneous enzymatic assay. Endotoxin was neutralized by the pre-treatment of TMCPM for 15 min. with polymyxin B (100 µg/mL). 

To further analyze the role of endotoxin content in TMCPM on the activity of CD4+ T cells we examined how the treatment of NIST and LAP with polymyxin B, a potent inhibitor of LPS activity, affects the Th1/Th2 and Treg/Th17 balance. To this end we analyzed the response of T lymphocytes, PBMC and T lymphocytes with monocytes added (10%) after stimulation with TMCPM treated with polymyxin B. The obtained results are presented in Figure 7 and Figure 8, respectively. 

These results clearly document that polymyxin B treatment, although almost completely eliminated LPS activity in NIST (0.1 EU/mL) and completely in LAP preparation (0.0 EU/mL) does not abrogate activation of Th1 and Th17 subsets in PBMC and in a co-culture of T cells with monocytes stimulated with such TMCPM. This data implicates, that in addition to relevance of organic compounds (including LPS), also the role of inorganic components of TMCPM in the activation of Th subsets. It is worth mentioning that treatment of TMCPM by polymyxin B itself caused an increase in IFN-γ production by T cells, observed especially in a positive control group (PMA+ionomycin), albeit with no statistical significance. 

### 3.6. Both Organic and Inorganic Components of TMCPM Might Be Responsible for Activation of Monocytes and Propagation of Inflammatory Response

Next, to check how monocytes respond to stimulation with TMCPM treated with polymyxin B we analyzed TNFα production and secretion by monocytes after stimulation with NIST and LAP, previously treated or not for LPS elimination. The data presented in Figure 9 clearly document that treatment of TMCPM with polymyxin B does not completely abolish activation of monocytes and production/secretion of pro-inflammatory TNFα, suggesting the important contribution in this phenomenon of inorganic components.

## 4. Discussion

The growing list of evidence has documented the connection between concentration of fine particulate matter in the air and initiation, development or exacerbation of allergies, inflammatory and autoimmune disorders [3,4]. In this report, we assessed the effect of two preparations of TMCPM, namely NIST and LAP, differing in content of organic compounds, on the activity of human CD4+ T cell subsets. This was determined by the expression of intracellular proteins (IFN-γ, IL-4, IL-17A, and Foxp3) characteristic for specific CD4+ T lymphocyte subpopulations (Th1, Th2, Th17, and Treg, respectively). In addition, the ratios of Th1/Th2 and Treg/Th17 cells after the treatment of PBMC with different concentrations of TMCPM were analyzed, as the optimal balance of Th1 to Th2 and Th17 to Treg is a key element responsible for maintaining homeostasis in the immune system. The obtained data suggest that exposition of PBMC to TMCPM induces strong polarization of CD4+ T cells into Th1 and Th17 subsets. This was shown by increase in the proportion of IFN-γ and IL-17A producing cells, with concomitant decrease of the level of Treg cells. In a similar study, Ma et al. also observed increased Th1 and Th17-like cytokine secretion after exposure of PBMC to particulate matter 2.5 µm [32]. The main difference between Ma’s and our study is the size and composition of the PM. Contrary to our study, where the standard reference urban air PM (SRM 1648a) was used, in the cited work the exact composition of the dust was not specified. Even though the final output of PBMC exposition to PM was the same—an increase in Th1 and Th17 immune response. It seems, that although the size and composition of the PM samples used in different studies may vary significantly, finally they lead to induction of the inflammatory reactions, albeit with different intensity. In keeping, it was documented, that the exposition of mice to ultrafine particles (less than 0.1 μm) elevates the frequency of Th1 cells in circulation and reduces the level of Treg cells in the lungs [33]. Moreover, Zhang et al. described the increased ratio of Th17/Treg after the exposition of rats to smoke, suggesting its potential role in pathogenesis of smoke inhalation-induced acute lung injury [34], while the results from similar study in mice link Th17/Treg cells imbalance with pathogenesis of cigarette induced emphysema [35]. 

It is worth mentioning that, although generally similar pattern of Th1 and Th17 cell activation have been observed for both preparations of TMCPM used in our study, we have noticed that NIST particles were much more potent in stimulation of T cells, as significantly higher production of IFN-γ and IL-17A was observed for the original NIST preparation compared to LAP. In keeping, recently Mikrut et al., using the same preparations of TMCPM showed that the organic fraction from NIST was mostly responsible for singlet oxygen photogeneration when compared with LAP particles [23]. In another study, Gawda et al. showed that the urban particulate matter SRM 1648a induced in vitro a dose-dependent production of pro-inflammatory cytokines (TNF-α, IL-6, IL-12p40) by murine macrophages [22]. Interestingly, under the same conditions, LAP particles were not able to stimulate macrophages, however the cells primed with NIST or LAP showed a strong pro-inflammatory response upon LPS challenge. These observations document important role of organic components content in a pro-inflammatory action of TMCPM. 

Of note, in our in vitro settings the described phenomenon was observed only when a whole population of PBMC had been used or purified monocytes exposed to TMCPM had been added to T lymphocytes, as the exposition of T lymphocytes only to TMCPM had no effect on the CD4+ subsets polarization. This implicates a crucial role of monocytes in activation of CD4+ T cells by TMCPM. Indeed, in our experimental settings monocytes stimulated with TMCPM produced high amounts of pro-inflammatory cytokines, depending on the organic compounds content in TMCPM preparations. Additionally, in Ma’s study, the increase in IFN-γ, IL-10, IL-17, and IL-21 production after the exposure to particulate matter 2.5 µm required the presence of cells of monocyte origin, as CD4+ and CD8+ T cells treated with PM2.5 in the absence of macrophages did not present higher IFN-γ, IL-10, or IL-21 expression [32]. Moreover, recently Castañeda et al. observed similar pattern, where activation of mouse dendritic cells after PM treatment was crucial for differentiation of naive CD4 T cells towards Th17 subset [36]. All together, these observations implicate that monocytes and cells of their origin, e.g., macrophages/dendritic cells are the primary cells responding to PM. In respect to the organic compounds, TLRs are natural candidates for their recognition and initiation of inflammatory response of monocytes/macrophages to PM. Several other studies [37,38,39,40,41] already suggested the involvement of endotoxin in promoting inflammation induced by PM. This was further supported by the effects of LPS inhibitors polymyxin B and LPS-binding protein on the reduction of pro-inflammatory activity of PM in vitro [42,43]. Although in our in vitro system endotoxin concentrations in TMCPM preparations correlated with pro-inflammatory cytokine secretion by monocytes, this effect was not completely abrogated by polymyxin B treatment. These data suggest that endotoxin, although present in TMCPM compositions, may not be solely responsible for a robust cell activation. We cannot exclude other, yet unknown, organic constituents within TMCPM, especially when several other insoluble or microbial components have been implicated in the increased macrophage responsiveness to PM [38,43]. However, our LAP preparation contained a very low amount of organic compounds, limiting their contribution in this phenomenon. It seems that endotoxin is partially responsible for pro-inflammatory potency of TMCPM and its effect is further enhanced by inorganic components, including, e.g., transition metals, triggering the Fenton’s reaction and generation of ROS (reactive oxygen intermediates) [44]. Actually, our unpublished data confirm this scenario in monocytes (Gałuszka et al, in preparation). 

In summary, our results indicate that treatment of human PBMC with TMCPM affects the balance of Th1/Th2 and Treg/Th17 cells, promoting activity of Th1 and Th17 subsets. This effect is strongly dependent on monocytes and the organic compounds, including LPS in TMCPM composition. Polymyxin B treatment, however, does not completely abrogate activity of TMCPM to monocytes, suggesting the role also for inorganic components of air pollutants. The obtained data suggest that polarization of Th cells toward Th1 and Th17 immune response is a potential mechanism promoting development and progression of allergy, and inflammatory and autoimmune disorders in humans after exposure to TMCPM. 

## 5. Conclusions

In conclusion, results of the present study indicated that in vitro exposure of PBMC to TMCPM stimulates the expression of IFN-γ and IL-17A and decreases expression of FoxP3, promoting pro-inflammatory activity of the Th1 and Th17 subsets. This effect depends on monocytes, organic (including LPS) and inorganic compounds content in TMCPM. These findings support the hypothesis that TMCPM may contribute to the development and exacerbation of allergies, and inflammatory and autoimmune disorders.

## Figures and Tables

**Figure 1 ijerph-17-01227-f001:**
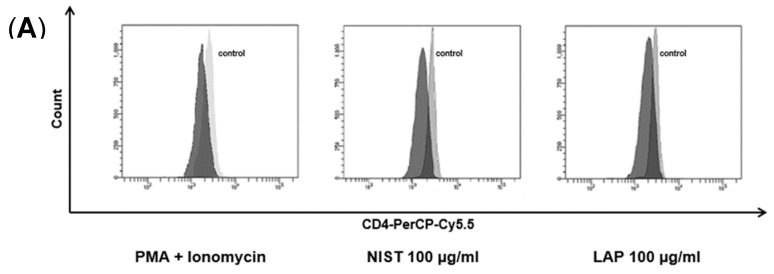
Effect of TMCPM (NIST or LAP) on the level of expression of CD4 on the surface of T cells and frequency of CD4+ T cells. PBMC were stimulated for 5 h with PMA + ionomycin or with NIST or LAP particles, stained with anti-CD4-PerCP-Cy5.5 conjugated mAb and analyzed by flow cytometry. (**A**) Histogram overlay shows fluorescence intensity (PerCP-Cy5.5) of CD4+ cells. Grey histogram—fluorescence intensity of CD4+ cells in a control culture (without stimulation), black histogram—fluorescence intensity of CD4+ cells in the samples stimulated either with PMA (50 ng/mL) + ionomycin (100 ng/mL) (left), 100 µg/mL LAP (middle) or 100 µg/mL NIST (right). Note a decreased expression of CD4 after stimulations of PBMC cells (decreased mean fluorescence intensity). Representative data from one out of thirteen experiments performed are shown. (**B**) Mean ± SEM percentage values of CD4 positive lymphocytes in control cultures (without stimulation) and in the groups stimulated either with PMA (50 ng/mL) + ionomycin (100 ng/mL) or LAP or NIST used at three different concentrations (1 µg/mL, 10 µg/mL, 100 µg/mL), calculated from data obtained from thirteen independent experiments (one-way ANOVA test).

**Figure 2 ijerph-17-01227-f002:**
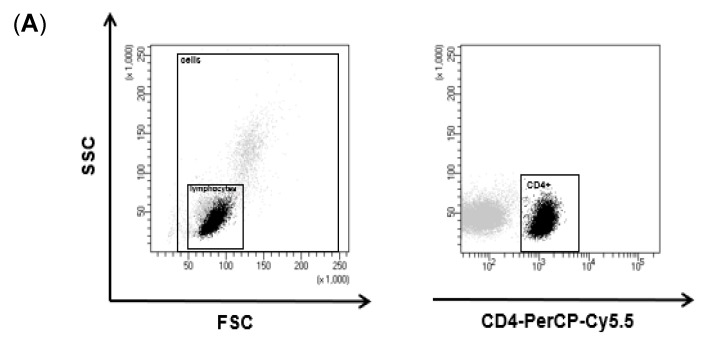
Flow cytometry analysis of intracellular proteins characteristic for specific Th subsets after exposition of PBMC to TMCPM. (**A**) Gating strategy for CD4+ T cell identification. Lymphocytes were defined according to FSC vs. SSC parameters from PBMC and then Th cells were gated according to CD4 expression. (**B**) Dot plots (IFN-γ-FITC vs. IL-4-APC; IL-17A-PE vs. FoxP3-APC) show CD4+ T cells positive for IFN-γ, IL-4, IL-17A and FoxP3 in unstimulated control, cells stimulated with PMA (50 ng/mL) + ionomycin (100 ng/mL) (positive control) or stimulated with NIST (100 µg/mL) or LAP (100 µg/mL). Shown are data from one representative experiment out of thirteen performed. The expression of intracellular proteins IFN-γ, IL-4, IL-17A and FoxP3 was analyzed by flow cytometry after the cells were staining with the cocktail of fluorescently conjugated monoclonal antibodies, specific for IFN-γ (Th1), IL-4 (Th2), IL-17A (Th17) and FoxP3 (Treg).

**Figure 3 ijerph-17-01227-f003:**
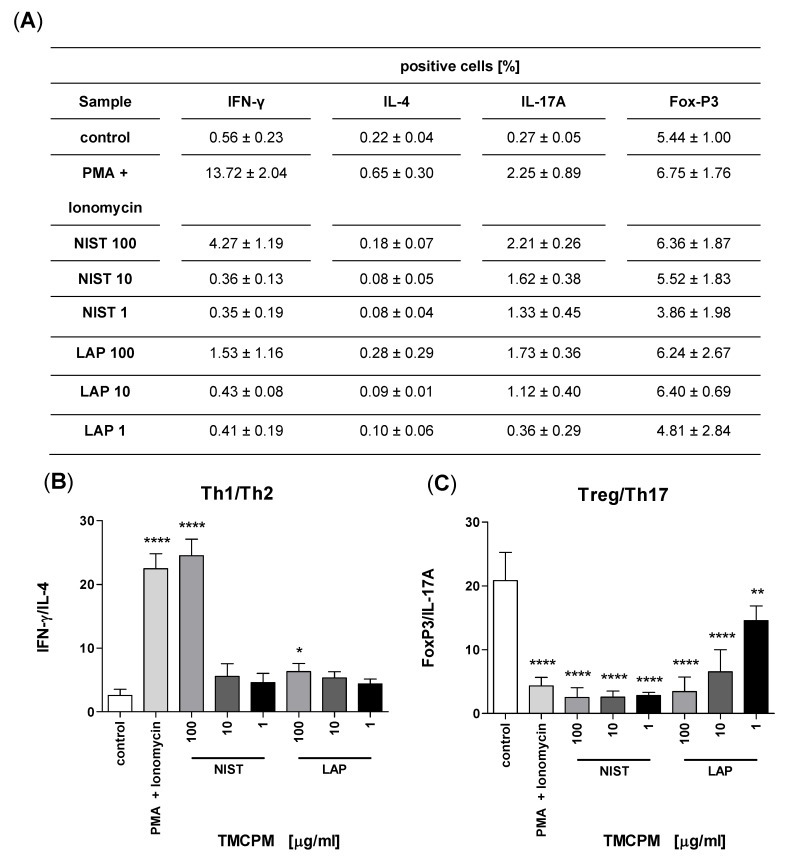
Effect of PBMC exposition to TMCPM (NIST and LAP) on the ratio of Th1/Th2 and Treg/Th17 T cells. PBMC were stimulated as described in Materials and methods and percentage of Th1, Th2, Th17, and Treg cells was quantified as the frequency of cells positive for IFN-γ, IL-4, IL-17A, and FoxP3, respectively (**A**). The ratio of the frequency (%) of IFN-γ positive to IL-4 positive CD4+ T cells (Th1/Th2) (**B**) and FoxP3 positive to IL-17A positive CD4+ T cells (Treg/Th17) (**C**) was calculated. Data are presented as mean ± SEM from twelve independent experiments. Differences between the groups were considered statistically significant at the following *p* values: * *p* < 0.1, ** *p* < 0.01, **** *p* < 0.0001.

**Figure 4 ijerph-17-01227-f004:**
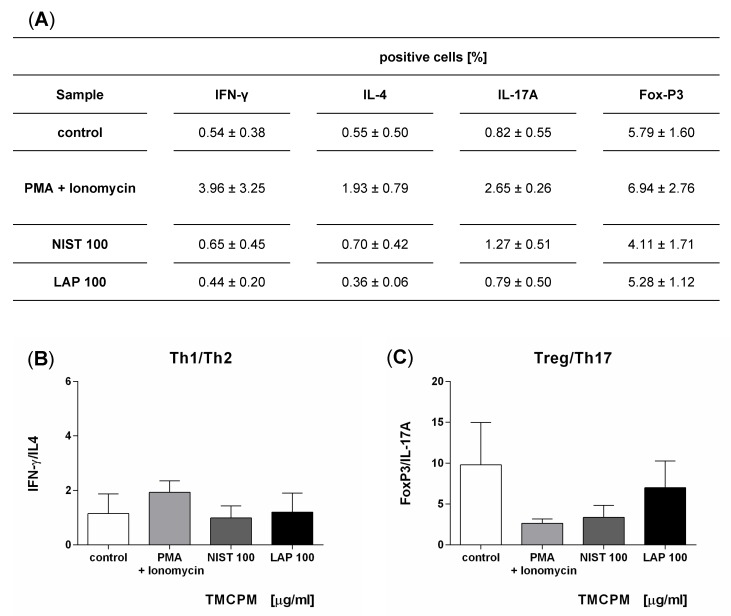
Effect of the exposition of T lymphocytes to TMCPM (NIST or LAP) on the Th1/Th2 and Treg/Th17 ratios. T lymphocytes were stimulated with PMA (50 ng/mL) + ionomycin (100 ng/mL) (positive control) or with NIST or LAP (100 µg/mL). Percentage of Th1, Th2, Th17, and Treg cells was quantified as the frequency of cells positive for IFN-γ, IL-4, IL-17A and FoxP3, respectively (**A**). The frequency ratio (%) of IFN-γ positive to IL-4 positive CD4+ T cells (Th1/Th2) (**B**) and FoxP3 positive to IL-17A positive CD4+ T cells (Treg/Th17) (**C**), was calculated. Data are presented as the mean ± SEM from three independent experiments performed.

**Figure 5 ijerph-17-01227-f005:**
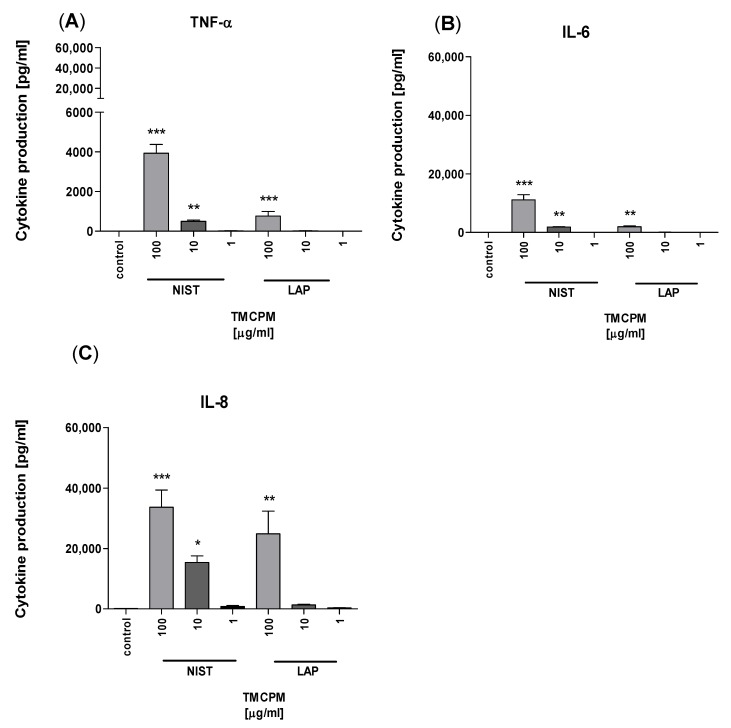
Effect of TMCPM (LAP or NIST) on cytokine production by purified human monocytes. Monocytes were stimulated with LAP or NIST at three different concentrations (1 µg/mL, 10 µg/mL, 100 µg/mL). Concentrations of TNF-α (**A**), IL-6 (**B**), and IL-8 (**C**) in culture supernatants were determined by CBA. Data are presented as mean ± SEM. The figure shows the results from three independent experiments. Differences between the groups were considered statistically significant at the following *p* values: * *p* < 0.1, ** *p* < 0.01, *** *p* < 0.001, **** *p* < 0.0001.

**Figure 6 ijerph-17-01227-f006:**
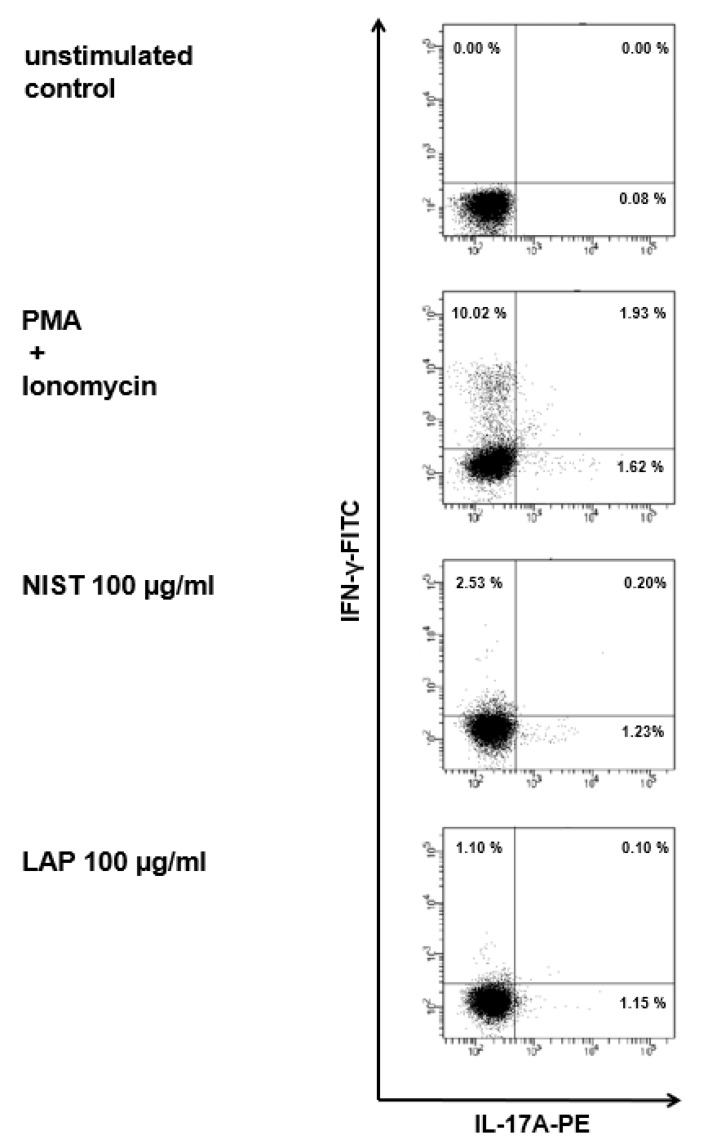
Flow cytometry analysis of Th1 and Th17 subsets after co-incubation of purified monocytes exposed to TMCPM with T lymphocytes. Dot plots (IFN-γ-FITC vs. IL-17A-PE) show CD4+ T cells positive for IFN-γ and IL-17A in unstimulated control, cells stimulated with PMA (50 ng/mL) + ionomycin (100 ng/mL) (positive control) and stimulated with NIST (100 µg/mL) or LAP (100 µg/mL). Shown are data from one representative experiment out of six performed. The expression of intracellular IFN-γ and IL-17A was analyzed by flow cytometry after the cells were stained with the cocktail of fluorescently conjugated monoclonal antibodies, specific for IFN-γ (Th1) and IL-17A (Th17).

**Figure 7 ijerph-17-01227-f007:**
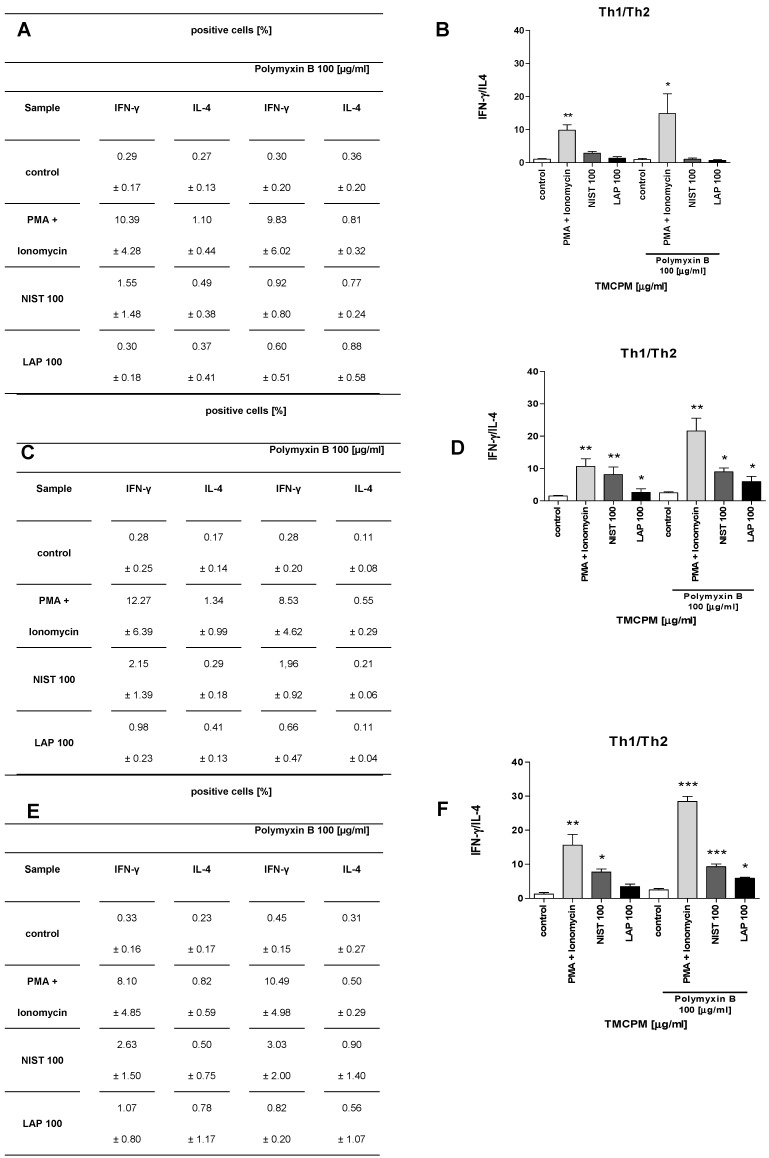
Effect of the exposition of T lymphocytes (**A**,**B**), PBMC (**C**,**D**) and T lymphocytes with monocytes added (**E**,**F**) to TMCPM (NIST or LAP) pre-treated with polymyxin B on the Th1/Th2 ratio. Cells were stimulated as described in “Materials and Methods” with PMA (50 ng/mL) + ionomycin (100 ng/mL) (positive control) or with NIST or LAP (100 µg/mL) pre-treated with polymyxin B (100 µg/mL). (**A**,**C**,**E**) Percentage of Th1, Th2, Th17, and Treg cells was quantified as the frequency of cells positive for IFN-γ, IL-4, IL-17A, and FoxP3, respectively. (**B**,**D**,**F**) The frequency ratio (%) of IFN-γ positive to IL-4 positive CD4+ T cells (Th1/Th2) was calculated. Data are presented as mean ± SEM from four independent experiments performed. Differences between the groups were considered statistically significant at the following *p* values: * *p* < 0.1, ** *p* < 0.01, *** *p* < 0.001.

**Figure 8 ijerph-17-01227-f008:**
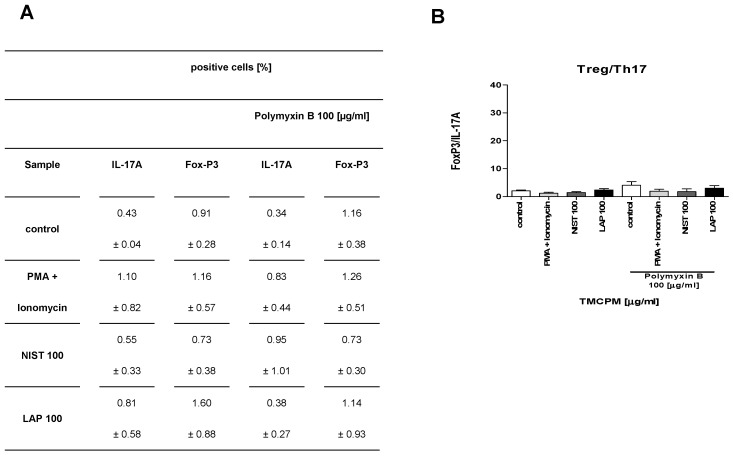
Effect of the exposition of T lymphocytes (**A**,**B**), PBMC (**C**,**D**) and T lymphocytes with monocytes added (**E**,**F**) to TMCPM (NIST or LAP) pre-treated with polymyxin B (100 µg/mL) on the Treg/Th17 ratio. Cells were stimulated as described in “Materials and Methods” with PMA (50 ng/mL) + ionomycin (100 ng/mL) (positive control) and with NIST or LAP (100 µg/mL) pre-treated with Polymyxin B (100 µg/mL). (**A**,**C**,**E**) Percentage of Th1, Th2, Th17, and Treg cells was quantified as the frequency of cells positive for IFN-γ, IL-4, IL-17A and FoxP3, respectively. (**B**,**D**,**F**) The ratio of the frequency (%) of FoxP3 positive to IL-17A positive CD4+ T cells (Treg/Th17) was calculated. The data are presented as mean ± SEM from four independent experiments performed. Differences between the groups were considered statistically significant at the following *p* values: * *p* < 0.1, ** *p* < 0.01, *** *p* < 0.001.

**Figure 9 ijerph-17-01227-f009:**
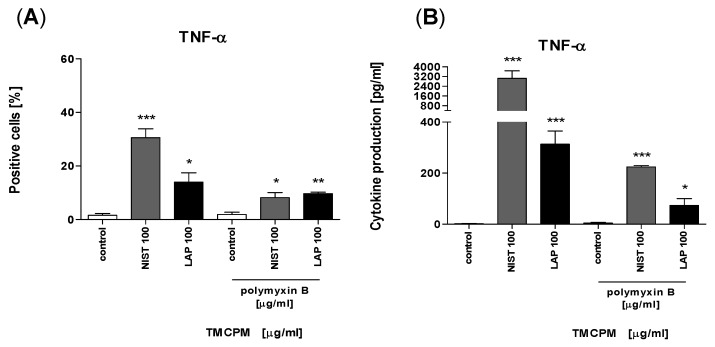
Effect of LPS inactivation in TMCPM (NIST and LAP) on TNFα production by purified human monocytes. Monocytes were stimulated for four hours with NIST or LAP (100 µg/mL) with or without pre-treatment with polymyxin B (100 µg/mL). Percentage values of TNF-α positive monocytes (**A**) was analyzed by flow cytometry after the cells were stained with fluorescently conjugated monoclonal antibodies. Concentration of TNF-α (**B**) in culture supernatants were determined by CBA. Data are presented as mean ± SEM. The figure shows the results from three independent experiments. Differences between the groups were considered statistically significant at the following *p* values: * *p* < 0.1, ** *p* < 0.01, *** *p* < 0.001.

**Table 1 ijerph-17-01227-t001:** Endotoxin concentration (EU/mL) in TMCPM (NIST and LAP) preparations with or without polymyxin B pre-treatment.

	Sample	LPS [EU/ml]		
				Polymyxin B 100 [µg/mL]
*E. coli*	[ng/mL]	LPS 0.5	5.50	0.00
TMCPM	[µg/mL]	NIST 100	1.75	0.10
		LAP 100	0.40	0.00

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
