# Peer review of "Transition Metal Containing Particulate Matter Promotes Th1 and Th17 Inflammatory Response by Monocyte Activation in Organic and Inorganic Compounds Dependent Manner"

_ijerph, 2020, doi:10.3390/ijerph17041227_

Round 1
Reviewer 1 Report
This version of the paper is significantly improved. The authors have experimentally addressed the main issue of the paper and, therefore, I have no further comments.
Author Response
We would like to thank this Reviewer for their valuable comments.
Reviewer 2 Report
The data presented here rely on the ratio, such as INF-y/IL4 and FoxP3/Th17, to show the effects of particle treatment. This somewhat loses the sight on the changes of each variable after treatment. The authors may use a table or any other to amend this. [Figs 7 and 8] It’s a common practice to include NIST or LAP without Polymyxin B treatment in the experiment, and present the results together with those from particles treated with Polymyxin B.Author Response
We would like to thank the Reviewers for their valuable comments regarding our manuscript entitled “Transition metal containing particulate matter promotes Th1 and Th17 inflammatory response by monocyte activation in organic and inorganic compounds dependent manner”.
Where appropriate, changes have been implemented in the revised manuscript (marked in red), also comments have been listed below.
The data presented here rely on the ratio, such as INF-y/IL4 and FoxP3/Th17, to show the effects of particle treatment. This somewhat loses the sight on the changes of each variable after treatment. The authors may use a table or any other to amend this.
Appropriate changes have been made in line with the Reviewer’s comments and the required data have been introduced into the Fig.3, Fig.4 as a new panel A (table) and into Fig.7 and Fig.8 as new panels A,C,E (tables). The figure legend was changed accordingly.
[Figs 7 and 8] It’s a common practice to include NIST or LAP without Polymyxin B treatment in the experiment, and present the results together with those from particles treated with Polymyxin B.
We thank the Reviewer for this comment. The required data on NIST and LAP activity without Polymyxin B treatment have been introduced to the respective figures (Fig.7 and 8) and the figure legend was changed accordingly. Moreover, a sentence clarifying these data was introduced in the body text (line 407-410).
We do hope that the Reviewer will find our answers satisfactory.
This manuscript is a resubmission of an earlier submission. The following is a list of the peer review reports and author responses from that submission.
Round 1
Reviewer 1 Report
This paper addresses the impact of TMCPM in the differentiation of different T cell phenotypes. While the model used is limited (PBMCs directly isolated from healthy humans are stimulated with TMCPM) the question is relevant and the hypothesis is well exposed and clear to the reader.
The data show that, TMCPM do impact the differentiation of Th cells and the authors guide the reader through these alterations throughout the paper. However, in the end of the paper it is shown that the impact of TMCPM in the differentiation is, in part, mediated by LPS. This iece of data is critical as the presence of LPS in the assays used in this paper, precludes any conclusion pertaining the effect of TMCPM. Therefore, it is of critical importance to perform these experiments with TMCP that are devoid of LPS to be able to draw any conclusion in what regards the effect of these particulate matter in the differentiation of T cells. Otherwise, it is not clear if it is TMCPM or LPS that is promoting Th differentiation. For example, one of the conclusions is that TMCPM are inflammatory and promote Th1 and Th17 differentiation. However, in the last figure TMCPM containing LPS induces less TNF and IL-1b that LPS alone suggesting that TMCPM are anti-inflammatory and inhibit TNF and IL-1b production.
Reviewer 2 Report
The specific work investigates whether transition metal containing particulate matter (TMCPM) has any effect on the activity of CD4+ T cell subsets. The issue of the special concern and all the new data are welcome to literature. The work is well written and can be accepted to its present form.
Minor changes needed. To be more specific, there are many points in the manuscript where the third person should be used instead of the first eg. line 12 we ; line 54 we..
line 143, 336-337, 349, 360, 217(our study), 325, 375, 387, 391. line 298: his... please add t.
line 140: why 5 hours; is it a normal procedure; is it according any protocal;
line 344: zhang et a., ..please add the date
Reviewer 3 Report
1. [Page 1, Line 20; Figures 6 and 7; Page 12, Line 294-5]. The authors claim that TMCPM promotes Th1 and Th17 inflammatory responses by monocytes. As shown in Figure 7, TMCPM stimulates the productions of TNF-alpha, IL-6 and IL8 in human purified monocytes. In page 12, the authors state that co-incubation of purified monocytes with lymphocytes in the presence of TMCPM promotes the Th1 and Th17 polarization of CD4+ cells. However, data are not shown (Line 294-5). Since the results in Figure 6 suggest that there is no Th polarization of lymphocytes by TMCPM, it is important to show the direct evidence that activated monocytes are required for the polarization of CD4+ cells to Th1 or Th17 cells (page 1, line 20). Please provide the data on the flow cytometry analysis of Th1 and Th17 polarization of CD4+ cells after co-incubation with monocytes in the presence of TMCPM, as a new Figure 8.
2. [Figure 1, Figure 2] In this in vitro study, CD4+ positive cells were not significantly increased after exposure to TMCPM (Figure 1). In figure 2, the authors attempted to use intracellular expression of FoxP3, IL17, IL4, and INF-gamma to characterize the specific Th subsets of Treg, Th17, Th2, and Th1, respectively. As discussed above, the authors stressed that Th polarization may require the activation of monocytes (Page 1, Line 20). Thus, it is important to include surface markers, such as CD25 and CD4, in determining Th polarization after exposure of PBMC to TMCPM. If the authors have included CD4 in the flow cytometry analysis, as stated in page 3 (line 102), please show the gating of CD4 in figure 2.
3. [Table 1, Figure 8] The endotoxin concentrations in LAP and NIST are 0.5 and 3.83 ng/mL, respectively (Table 1). However, TNF-alpha and IL-1beta productions of LAP and NIST treated monocytes were significantly less than those in monocytes treated with purified LPS at 0.5 and 4ng/mL, respectively. If the authors would conclude that proinflammatory response or activation of monocytes depends on the organic contents, including LPS endotoxin content, in TMCPM (page 1, line 21), the use of a LPS inhibitor or the heat inactivation of endotoxin would serve the purpose well. In addition, figure 8 only shows the results of one representative experiment out of three performed (page 14, line 320-1). If there are three experiments performed, please include all three in the current figure 8.